# Differential Proliferation Effect of the Newly Synthesized Valine, Tyrosine and Tryptophan–Naphthoquinones in Immortal and Tumorigenic Cervical Cell Lines

**DOI:** 10.3390/molecules25092058

**Published:** 2020-04-28

**Authors:** Sergio Córdova-Rivas, Jorge Gustavo Araujo-Huitrado, Ernesto Rivera-Avalos, Ismailia L. Escalante-García, Sergio M. Durón-Torres, Yamilé López-Hernández, Hiram Hernández-López, Lluvia López, Denisse de Loera, Jesús Adrián López

**Affiliations:** 1Laboratorio de microRNAs y Cáncer, Unidad Académica de Ciencias Biológicas, Universidad Autónoma de Zacatecas, Zacatecas 98068, Mexico; 2School of Chemistry, Universidad Autónoma de San Luis Potosí, San Luis Potosí 78210, Mexico; 3Unidad Académica de Ciencias Químicas, Universidad Autónoma de Zacatecas, Zacatecas 98160, Mexico; 4CONACYT, Laboratorio de Metabolómica y Proteómica, Unidad Académica de Ciencias Biológicas, Universidad Autónoma de Zacatecas, Zacatecas 98068, Mexico; 5Instituto de Investigación de Zonas Desérticas, Universidad Autónoma de San Luis Potosí, San Luis Potosí 78377, Mexico

**Keywords:** naphthoquinone, amino acids, alternative methods, microwave, ultrasound, anticancer

## Abstract

We previously showed that microwave assisted synthesis is the best method for the synthesis of naphthoquinone amino acid and chloride-naphthoquinone amino acid derivatives by a complete evaluation of reaction conditions such as stoichiometry, bases, and pH influence. Following the same strategy, we synthesized chloride and non-chloride tyrosine, valine, and tryptophan-naphthoquinones achieving 85–95%, 80–92%, and 91–95% yields, respectively. The cyclic voltammetry profiles showed that both series of naphthoquinone amino acid derivatives mainly display one redox reaction process. Overall, chloride naphthoquinone amino acid derivatives exhibited redox potential values (E_1/2_) more positive than non-chloride compounds. The six newly synthesized compounds were tested in HPV positive and negative as well as in immortal and tumorigenic cell lines to observe the effects in different cellular context simulating precancerous and cancerous status. A dose-response was achieved to determine the IC50 of six newly synthesized compounds in SiHa (Tumorigenic and HPV16 positive), CaLo (Tumorigenic and HPV18 positive), C33-A (Tumorigenic and HPV negative) and HaCaT (Keratinocytes immortal HPV negative) cell lines. Non-chloride tryptophan-naphthoquinone (3c) and chloride tyrosine-naphthoquine (4a) effects were more potent in tumorigenic SiHa, CaLo, and C33-A cells with respect to non-tumorigenic HaCaT cells. Interestingly, there seems to be a differential effect in non-chloride and chloride naphthoquinone amino acid derivatives in tumorigenic versus non tumorigenic cells. Considering all naphthoquinone amino acid derivatives that our group synthesized, it seems that hydrophobic and aromatic amino acids have the greatest effect on cell proliferation inhibition. These results show promising compounds for cervical cancer treatment.

## 1. Introduction

Naphthoquinones (NQ) are a group of natural and synthetic compounds known for their antibacterial, antifungal, anticancer, antimalarial, and antiviral effects, among others [1,2]. Several studies have reported that naphthoquinones with different substituents like paclitaxel, esters, metals, furans, carbazoles, or inclusive with carbohydrates in their structure, among others, inhibit cell proliferation [3,4,5]. These properties are principally attributed to the oxidant-reductive characteristics of naphthoquinones, which allow the generation of dianions or semiquinone radicals. In the last decades, some publications have been dedicated to the finding of an explanation for the formation of these intermediates in the synthetic mechanism, and their properties that have produced different compounds with a plethora of applications of biological importance and effects that involve intra- or intermolecular interactions. One of the principal effects of these compounds is the generation of reactive oxygen species (ROS), producing cytotoxicity in different cell lines [6,7,8,9,10,11]. ROS generation with naphthoquinones represents a challenge in the design of new active compounds, principally with anticancer effects. In this context, the addition/substitution on naphthoquinone moiety by atoms or groups such as flour, oxygen, or amine can be modulated by redox properties, decreasing the toxicity levels and maintaining or potentiating the biological effect [12,13,14].

Cell proliferation inhibition could be achieved by the induction of apoptosis, topoisomerase II-α inhibition, and ROS generation, among others [3]. Furthermore, it has been shown that β-lapachone inhibits epidermal growth factor (pEGFR), protein kinase B (pAKT/PKB), glycogen synthase kinase (pGsk-3β), cyclin D1, and cyclooxygenase-2 (COX-2) protein expression in a dose-dependent manner [15]. Naphthoquinone regulation depends on the composition of its substituents, as well as targets and cellular components that could be inhibited and/or activated in cancer.

Cervical cancer is the second most common cancer in women worldwide, with an estimated global incidence of 470000 new cases and over 200000 deaths per year [16]. High-risk human papillomavirus (HPV) infection has been associated with the development of cervical, penile, and anal cancers. Although high-risk HPV expression appears necessary to immortalize and stimulate proliferation in human keratinocytes [17], it is not sufficient to produce a fully transformed phenotype suggesting that other genetic changes participate in the development of cervical cancer [18]. Several naphthoquinones have been produced; however, they usually have non-desirable effects in cancer therapy use. Therefore, the production of new compounds by alternative methods is imperative for cancer treatment.

In this regard, the synthesis and biological activities of some naphthoquinone–amino acids have been reported in several publications. Nonetheless, all of these methodologies present different drawbacks, like low yields and/or long reaction times [19,20,21,22,23]. To our knowledge, the synthesis of naphthoquinone–amino acid derivatives has not been reported using microwave and ultrasound irradiation until our previous work [24], which offer diverse advantages to conventional synthesis.

Our research group synthesized naphthoquinone amino acid derivatives to study their potential antitumorigenic activity against cervix and breast cancer cell lines. The compounds studied showed an inhibition grade ranked from 85% to 40% in SiHa cells, and 90% to 30% in MCF-7 cells. Surprisingly, chloride naphthoquinone amino acid derivatives possess a selective activity against breast cancer cell lines, on the other hand, without chlorine atom in their structure the naphtoquinone amino acid derivatives presented selectivity against cervix cancer cell lines [24]. To complete the study of naphthoquinone amino acid derivatives against cancer cell lines, in this paper, we present the synthesis of valine, tyrosine and tryptophan 1,4-naphthoquinone (3a–c), and chloro-naphthoquinone (4a–c) derivatives by microwave (MAS) and irradiation, and their effect in the cervical cancer cell lines HPV-positive SiHa and CaLo, HPV-negative C33-A and the keratinocytes non-cancerous HPV negative cell line, HaCaT. The incorporation of amino acids to naphthoquinones could enhance their cytotoxicity capacity as well as their specificity achieving its potential use as antitumorigenic compounds.

## 2. Results

### 2.1. Chemistry

Figure 1 shows the general method to prepare 3a–c and 4a–c derivatives, and in Table 1 are present the reaction conditions used based on the optimized conditions previously reported [24].

### 2.2. Electrochemical Studies by Cyclic Voltammetry

Cyclic voltammetry (CV) curves of naphthoquinone–amino acid (3a–c) and chloride–naphthoquinone–amino acids (4a–c) derivatives in 0.1M TBABF_4_/DMSO at 100 mV s^−1^ are reported in Figure 2A,B, respectively. Tyrosine, valine, and tryptophan naphthoquinone derivatives, without and with the chloride moiety, exhibit a similar electrochemical response to that for naphthoquinone derivatives modified with alanine, phenylalanine, methionine, glycine and asparagine as previously reported by our group [24]. One redox reaction is identified in the cathodic potential region at half-wave potentials values (E_1/2_) near to −1.2 V and −1.1 V for the naphthoquinone (3a–c) and chloride–naphthoquinone (4a–c) derivatives, respectively. Additionally, several irreversible oxidation processes are also observed at more positive potential values, >−0.75 V vs. Ag/Ag^+^ for 3a–c and 4a–c compounds (Figure 2). The cathodic waves at potentials near to the redox reaction are related to the electrochemical reduction of these oxidation reactions since voltammetry measurements in a shorter potential range, from −0.5 V to −2.0 V vs. Ag/Ag^+^, did not display those cathodic waves. The redox reaction at E_1/2_ is associated with the reduction of the quinone moiety (Q) of compounds 3a–c or 4a–c by a single step two-electron transference process to yield the quinone dianion (Q^2−^) (Figure 3, reaction (i)) [25,26,27]. To recall, the electrochemical reduction of typical quinone moiety (Q) in nonaqueous media exhibits two successive one electron-transfer steps forming the semiquinone radical (Q^•−^) and quinone dianion (Q^2−^) (Figure 3, reaction (ii). Thus, two redox processes at different half-wave potentials are expected [24,25,26,27]. In this study, it is then suggested that the reduction mechanism of the quinone moiety (Q) in 3a–c and 4a–c is influenced by electronic and steric effects due to the amino acid substituent groups (tyrosine, valine and tryptophan); as consequence, only one well-defined redox process is observed (Figure 2). Possibly, electronic interference of the amine group between the quinone ring and the amino acid substituent, as well as strong intramolecular hydrogen bonding help to stabilize the quinone moiety (Q) or the radicals intermediates (Q^•−^ and Q^2−^) [25,27,28,29,30].

The electrochemical parameters of the redox reaction were evaluated by cyclic voltammetry at different scan rates and are summarized in Table 2. First of all, the redox reaction of 3a–c and 4a–c is a diffusion-controlled process according to a linear relationship of the cathodic peak current density (i_pc_) with the square root of the scan rate (i_pc_ vs. v^1/2^), thus indicating fast chemical kinetics. The potential peak separation values (ΔE_p_) are quite similar among compounds 3a–c or 4a–c. Here, it is worth noting that the redox reaction for the naphthoquinone with tryptophan (3c) and chloride-naphthoquinone with valine (4b) exhibit the lowest ΔE_p_, 0.11 V vs. Ag/Ag^+^, therefore, faster kinetics are occurring. Additionally, compounds 3a–c exhibit good reversibility according to the ratio of the anodic peak current to cathodic peak current (i_pa_/i_pc_) which is close to one in value as described for reversible systems [31]. Chloride-naphthoquinone amino acid derivatives, 4a–c, behave quite irreversible as compared to 3a–c compounds according to i_pa_/i_pc_ values reported in Table 2. The E_1/2_ for the redox reaction was in the potential range of −1.21 V to −1.24 V and −1.10 V to −1.12 V for 3a–c and 4a–c, respectively, as reported in Table 2. Note that E_1/2_ for 4a–c compounds shifts toward anodic potentials by at least 110 mV as compared to the E_1/2_ registered for non-chloride naphthoquinone amino acid derivatives, 3a–c. The potential shift is related to the electron-attracting capacity of the chloride substituent in ortho-position that facilitate the electron transfer and the electrochemical reduction of the quinone moiety [28,32]. Here, the potential shift is related to inductive and resonance effects of the halogen substituent at this position [28]. These results are consistent with those previously reported by our group for naphthoquinone amino acid derivatives modified with alanine, phenylalanine, methionine, glycine and asparagine [24]. Nonetheless, the naphthoquinone amino acid derivatives reported before demonstrated higher reversibility and slightly lower kinetics.

Besides the redox process of the quinone moiety, naphthoquinone amino acid derivatives show two or more irreversible oxidation waves at anodic potentials between −0.75 V and 0.75 V vs. Ag/Ag^+^ as mentioned above (Figure 2). Particularly, a well-defined oxidation potential peak is identified at 0.60 V and 0.45 V for naphthoquinones modified with valine (3b) and tryptophan (3c) as shown in Figure 2A, respectively; as well as for the corresponding amino-chloride-naphthoquinones derivatives, valine (4b) and tryptophan (4c), at 0.57 V and 0.46 V, respectively (Figure 2B). Compounds 3a and 4a exhibit an ill-defined oxidation wave at potential values higher than 0.5 V as shown in Figure 2. Additionally, chloride naphthoquinone amino acid derivatives display a noticeable anodic peak at −0.33V or −0.36 V as shown in Figure 2B for 4a or 4b,c, respectively. Naphthoquinone derivative with valine (3b) also exhibits an anodic peak at −0.2 V meanwhile 3a and 3c compounds display several overlapping oxidation processes in the potential range of −0.75 V to 0.25 V (Figure 2A). All of these irreversible anodic processes are related to oxidation reactions by electron dislocation of the amino acid substituent group which could allow interaction with DNA, RNA and proteins in the cells affecting their proliferation.

### 2.3. Effect of Naphthoquinone Amino Acid Derivatives in Cell Proliferation of Tumorigenic Cervical Cancer Cell Lines and Immortal Cell Line

A desired feature of a new potential therapeutic agent is its differential effect between pre-cancerous and cancerous status, therefore, we tested two principal characteristics: 1) status of cells regarding the development toward cancer (immortal versus tumorigenic) and 2) HPV content (negative or positive). Immortal and HPV negative cell line HaCaT cells was selected as control for both immortality status and HPV content. SiHa (HPV16 positive), CaLo (HPV positive), and C33-A (HPV negative) cells were evaluated considering their tumorigenic status and HPV content. In order to evaluate the effect of the naphthoquine amino acid derivatives in immortal and tumorigenic cells with HPV positive and negative status we performed a dose-response analysis. The forth cells were treated with 6.25, 12.5, 25, 5, 100, and 200 μM of naphthoquinone amino acid derivatives 3a–c and 4a–c. Compounds with tyrosine (3a), valine (3b), and tryptophan (3c) substituents showed a dose-dependent proliferation inhibition in all the cells tested, Figure 4A–D. IC50 of compound 3a was 40.98, 55.67, 61.60, and 81.08 for SiHa, HaCaT, CaLo and C33-A cells, respectively. A different proliferation inhibition was exerted by 3b compound, being SiHa the most affected, followed by HaCaT, C33-A, and CaLo. While proliferation inhibition was stronger in C33-A cells followed by CaLo, SiHa and HaCaT after 3c compound treatment, Table 3. The results analysis suggests that HPV status in cell lines doesn’t seem to be important in the compounds exerted effects. By contrast the immortal/tumorigenic status appears to be an important cell feature for non-chloride-tryptophan-naphthoquinone (3c) compound exerted effect. Interestingly, a similar behavior in proliferation inhibition was observed by chloride-tyrosine-naphthoquinone (4a), Figure 4 and Table 3. It is worth highlighting the strong difference observed between non-chloride (3a–c) and chloride compounds (4a–c), remarkably a difference of more than one order of magnitude with respect to the proliferation inhibition exerted by chloride compounds, as shown in Figure 4. It is of relevance to mention that almost all the compounds exerted lower proliferation inhibition over HaCaT cells in comparison to SiHa and C33-A cells, however, CaLo and HaCaT cells showed no important proliferation inhibition difference or even CaLo cells were less affected (3a,b), as shown in Figure 4 and Table 3. Finally, based on the low effect exerted on proliferation inhibition in HaCaT cells, which represent a closer resemblance to normal cells, we suggest 3c and 4a compounds are interesting candidates for further studies regarding cervical cancer therapeutic potential.

## 3. Discussion

The naphthoquinone–amino acid derivatives have been previously reported [3,5] and our group performed an extensive analysis about the reactivity of the equivalents of each amino acid used in the reactions, the dependence of the pH and the optimized reactions under ultrasonic and microwave irradiation reducing the reaction times from days or hours to only a few minutes to improve their production [24]. In this work tyrosine, valine and tryptophan naphthoquinone derivatives were synthesized obtaining excellent yields, 80–95%. The results show that chloride naphthoquinone derivatives yield increased using a weak base (TEA). On the other hand, to improve the yields in naphthoquinone derivatives, a strong base was needed (KOH), and this can be explained by the presence of a good releasing group in 2,3-dichloronaphthoquinone favoring the amino acid addition.

The overall electrochemical pathway of naphthoquinone amino acid derivatives is anticipated to be complicated since both the reduction of naphthoquinone moiety and oxidation of amino acid substituents are contributing to electron-transfer reactions as reported above. Although a thorough understanding of the electron-transfer pathway is not considered in this work, the electrochemical studies of naphthoquinone amino acid derivatives give an insight of possible biological reactions concerning electron transfer processes that could occur at cellular level. Furthermore, electrochemical parameters (Table 2) calculated by cyclic voltammetry are considered to predict the biological effect of newly synthesized compounds as reported elsewhere [33,34]. However, it is worth mentioning that a direct correlation of the electrochemical parameters with biological activity can be compromised due to the complexity of biomedical chemistry or, in such a case, to the complexity of the physiological environment in living systems as compared to the media used for the electrochemical studies. In this context, the redox potential or half-wave potential (E_1/2_) is an electrochemical parameter that has been generally used to assess the biological activity of naphthoquinone compounds [33,34,35]. Naphthoquinones with more positive E_1/2_ values are associated with higher biological effects. This is because a spontaneous reduction of the compounds occurs, and thus the generation of reactive oxygen species (ROS) is presumably facilitated. Recall that the main mechanism of action for naphthoquinones is the generation of ROS. According to this, chloride-naphthoquinone amino acid derivatives (4a–c) in this work are expected to present higher biological activity than the naphthoquinone derivatives without the chloride moiety (3a–c) as the E_1/2_ values reported in Table 2. The E_1/2_ values for the redox reaction of chloride-naphthoquinone amino acid derivatives (4a–c) were found in the potential range of −1.10 V to −1.12 V meanwhile non-chloride naphthoquinone amino acid derivatives (3a–c) exhibited values in the potential range of −1.21 V to −1.24 V. E_1/2_ for chloride-naphthoquinone amino acid derivatives is at least 110 mV more positive than non-chloride naphthoquinone amino acid derivatives (3a–c). Similar results were reported by our group for chloride naphthoquinone amino acids derivatives modified with alanine, phenylalanine, methionine, glycine, and asparagine, suggesting that the redox reaction, and as a consequence, ROS production could occur in a similar manner to the naphthoquinone compounds reported in the present study [24].

For amino acid substituent groups, the relationship between the electrochemical behavior and biological activity is unknown. Thus, further studies are needed for better understanding of the action mechanism. However, the electrochemical behavior related to the oxidation of tyrosine, valine, and tryptophan in the naphthoquinone compounds may contribute in some extent to antiproliferative activity. For instance, some authors indicate that the intensity of the current density due to electrochemical oxidation depends on how the amino acids adsorb or lay at the electrode surface, that is, if the electro-active amino acid substituents are externally exposed or hidden by the molecule structure (i.e., proteins, DNA, RNA molecules) to ease the electron-transfer process [36,37]. However, Dourado et al. [38] mentioned that the oxidation of amino acids is observed in the same potential region and presents similar electrochemical behavior either tested free or linked to other compounds or molecules. In this study, note that all naphthoquinone derivatives, 3a–c and 4a–c, exhibit oxidation processes at similar potential range but chloride-naphthoquinone amino acid derivatives display well-defined oxidation peaks with high-current density response. This behavior is also attributed to the influence of the chloride with the amino acid substituent as described previously for the reduction of the quinone moiety. In fact, it is suggested that chloride-naphthoquinone amino acid derivatives (4a–c) are easily-adsorbed on the electrode surface enhancing the electrochemical oxidation reaction of amino acid substituents. Thus, it is clear that chloride substituent group improves the electrochemical activity of the naphthoquinone amino acid derivatives, and consequently, they could have an important antiproliferative effect in combination with the amino acid substituent group, possibly facilitating the drug delivery into the cell. However, further studies are needed to get insights into the mechanism of drug metabolism or detoxification when amino acid substituents are involved.

Recently it has been shown a proliferation inhibition effect of naphthoquinone amino acid derivatives in different cell lines. Alanine, methionine, phenylalanine, glycine, and asparagine-naphthoquinones were synthesized showing inhibition ranked from ~40% to 85% and 30% to 90% in SiHa and MCF-7 cells, respectively. Interestingly, cervix cell line, SiHa, responded better to non-chloride while breast cell line MCF-7 did to chloride compounds [24]. Nevertheless, this observation was weak since it was only seen in one cell type from each type of cancer. However, in this study we included HPV18, HPV16 positive and HPV negative cervical cancer cell lines to further clarify our previous observation and assure whether cervical cancer cell lines better respond to non-chloride than to chloride amino acid-naphthoquinones. Additionally, cell lines with different levels of cancer development were considered to investigated the response to the compounds. Interestingly, in the present study it was appreciated a difference in HPV18, HPV16, and HPV negative cell types preferentially responding to chloride than to non-chloride compounds. Among non-chloride compounds, tryptophan-naphthoquinone (3c) exerted the lesser effect in proliferation inhibition in HaCaT cells (40.82 µM) and an ascendant effect in C33-A (IC50 = 21.36 µM), CaLo (IC50 = 25.20 µM) and SiHa (IC50 = 28.8 µM) cell lines. The difference in proliferation inhibition with 3c treatment between immortal and tumorigenic cells was near double, being an important feature regarding its potential use as cancer therapy drug. On the other hand, it could be observed that chloride naphthoquinone amino acid derivatives exerted a proliferation inhibition with more than one magnitude order with respect to non-chloride compounds. These observations are also consistent with the electrochemical results obtained in this work since chloride naphthoquinone amino acid derivatives (4a–c) were expected to present higher cytotoxicity effects than non-chloride naphthoquinone amino acids derivatives (3a–c). Therefore, a direct correlation of the electrochemical parameters for each chloride compound with the antiproliferative effect is suggested, however, it must be noted that further studies are needed. Remarkably, comparing chloride compounds between them, tryptophan-naphthoquinone (4c) has almost the most potent effect in all cell lines, except for C33-A. Interestingly, Tyrosine-naphthoquinone (4a) presented the lesser effect in immortal cell line HaCaT (IC50 = 9.882 µM) compared with CaLo (IC50 = 7.028 µM), SiHa (IC50 = 6.830 µM) and C33-A(IC50 = 0.001577 µM) suggesting it is a potential therapy candidate. However, based on the severe effect exerted on cellular proliferation inhibition they should be proved in animal models to assay risk-benefit. It should be noted that the differential effect on each cell type may obey to the different cell lines origin and genetic background, which is going to be reflected in cell proliferation inhibition exerted by naphthoquinone amino acid derivatives. For example, HPV18 positive CaLo cell line showed resistance (IC50 = 81.08 µM) to 3a treatment opposite to HPV16 positive SiHa cells (IC50 = 40.98 µM), albeit a medium effect is recorded in immortal HPV negative HaCaT (IC50 = 61.6 µM) and C33-A (IC50 = 55.64 µM) HPV negative cells. Despite structural differences between (3a), (3b), and (3c) compounds, all naphthoquinone amino acid derivatives present considerable levels of proliferation inhibition in all cells tested. The percentage inhibition obtained in our work was similar to other compounds. For example, in a study with the ileocecal adenocarcinoma cell line HCT-8, 86%, 84% and 59% of proliferation inhibition was shown with phenylalanine, alanine, and proline naphthoquinone derivatives. In the breast cancer cell line MDAMB-435 proliferation inhibition was of 100%, 100%, and 36%, while in human multiforme cell line SF-295, 86%, 83%, and 59% of inhibition was observed, respectively [20]. Interestingly, de Moraes et al. [20] added three complete amino acids to naphthoquinone structure showing similar cytotoxicity to doxorubicin with an IC50 of 0.48 in MDAMB-435 cell line. The effect of glycine, alanine, and phenylalanine derivatives synthesized by de Moraes et al. [20] was similar to our results, highlighting their potential as cytotoxicity compounds in cancer. Marastoni et al. [39], incorporated leucine, asparagine, phenylalanine and serine into naphthoquinone with diamine alkyl spacers inhibiting three subunits of proteasome linked to proliferation inhibition of the breast cancer cell line MDA and ovarian cancer cell line A2780 in the range of 100 μM similar to the non-chloride compounds concentration used in this study. Remarkably, it should be noted that the chloride naphthoquine amino acids derivatives reached the nM concentration. Additionally, in Marastoni et al.’s work, the compounds effect was evaluated in 15 × 10^3^ cells, while in our study the number of cells used was of 40 × 10^3^. The molecules per cell that we used in our study were fewer than Marastoni et al.′s study, suggesting that our compounds are more potent than the ones they tested. Interestingly, in this work, it was shown that chloride naphthoquinone amino acid derivatives have similar effect to Taxol, a common drug used in cervical cancer treatment [40]. Remarkably, the chloride naphthoquinone amino acid derivatives have almost double the effect of Taxol in SiHa and C33-A cells.

Amino acid naphthoquinone derivatives performed different effects on cell proliferation of cell lines, therefore, it could be theorized that each particular effect observed may be dependent on hydropathy index and protein occurrence of the amino acids. The tyrosine, valine, and tryptophan amino acids have −1.3, 4.2, and −0.9 hydropathy index, respectively. However, the non-chloride (3c) and chloride tryptophan-naphthoquinone (4c) recorded the most potent effect in tested cells, except in C33-A cells which was severely inhibited by valine-naphthoquinone (4b). However, comparing tyrosine and tryptophan synthesized in this work versus glycine and asparagine-naphthoquinone derivatives in previous work [24] it seems probable that hydropathy index could participate in the compounds effect recorder. Glycine and asparagine possess the negative hydropathy index −0.4 and −3.5, respectively. Nevertheless, it should be noted that asparagine hydropathy index is more negative that glycine’s hydropathy index, even though it is more abundant in proteins than Asparagine, existing 7.2 and 5.1 molecules per protein, respectively [24]. In contrast, it should be noted that valine hydropathy index is out of range versus glycine, asparagine, tryptophan, and tyrosine without taking into account the changes that the amino acid could have when it is part of a complex. The amino acids occurrence in proteins for tyrosine, valine and tryptophan is 3.2, 6.6, and 1.4, respectively. Since valine and tyrosine-naphthoquinones present similar effects, their occurrence in proteins is not determinant. In contrast, compound size seems not to be an important feature since it was shown valine-naphtoquinone only exerted a potent effect in C33-A cells. Even though, it has been demonstrated that amino acid size is important in the β-turn folding of some proteins therefore the substitution and/or interaction with amino acid-naphthoquinones could destabilize the proteins structure affecting their function. Additionally, it must be considered that tyrosine is involved in several cell signaling pathways and a possible competition with endogenous tyrosine could occur. The occurrence of tryptophan in proteins is the lowest among the essential amino acids, however it is present in a family of proteins called GW-182 that participate in miRNA-mediated repression [41], therefore, the interruption or competition with GW-182 protein synthesis could affect several cellular processes like proliferation, apoptosis, migration and invasion among others. Nevertheless, this hypothesis needs several experimental studies to be addressed. In contrast, chloride tyrosine, valine and triptophane-naphthoquinones (4a,b,c) selectively show antitumorigenic properties versus non-chloride compounds in cervix cancer cell lines, positioning them as promising compounds for specific cancer treatment.

## 4. Materials and Methods

### 4.1. General

Commercially supplied 1,4-naphthoquinone, 2,3-dichloronaphthoquinone, tyrosine, valine and tryptophan were used for the synthesis without further purification. ^1^H and ^13^C Nucleus Magnetic Resonance (NMR) spectra were recorded on a Bruker Avance III 400 MHz spectrometer (^1^H at 400, ^13^C at 101 MHz, Silbestreifen, Rheinbstetten, Germany). The spectra were acquired from solution in methanol-d_4_ at room temperature, TMS as internal reference, the chemical shifts (δ) are expressed in part per million (ppm) and the coupling constants (*J*) in Hz. High-resolution mass spectra (HRMS) were measured with a Jeol JMS-AccuTOF through DART (Direct Analysis in Real Time, Peabody, MA, USA) and by ESI (Electrospray Ionization) in an Agilent 6200 Series TOF (Santa Clara, CA, USA) and 6500 Series Q-TOF LC/MS System (Santa Clara, CA, USA). Infrared spectra were recorded on a Thermo Scientific NICOLET iS10 with ATR dispositive (SMART iTR, Madison, WI. USA). Melting points were determined using a Bicote-Stuart SMO 10 apparatus (Stone, Staffordshire, UK) and were uncorrected. Microwave Assisted Synthesis (MAS) was performed on a CEM Mars 6 oven with a carrousel device (Matthews, NC, USA). Ultrasound Assisted Synthesis (UAS) was carried out in an Autoscience Ultrasonic cleaner-AS2060B with 60 Watts of power (Lewisville, TX, USA). TLC was performed using silica gel 60 PF_254_ containing gypsum (Merck, Darmstadt, Germany). The isolated reaction products were found to be >95% purity by NMR analysis.

### 4.2. General Procedure for the Synthesis of Compounds 3a–c and 4a–c

The obtention and purification of the compounds were carried out under the methodology described by [24]. Briefly, the solution of respective amino acid (1.5–2.5 mmol) in dioxane/water (4:1, 15 mL for 3a–c derivatives or 20 mL for 4a–c derivatives), was basified with TEA or KOH (Sigma-Aldrich, St. Louis, MO, USA) aq. (pH 9–10) and activated under microwave irradiation under 110 °C, 250 W for a time of 10 min (Mars 6, CEM, ISASA Latam, S.A. Alajuela, Costa Rica). Then a solution of naphthoquinone (Sigma-Aldrich, St. Louis, MO, USA) 5 mL (1a) or 2,3-dichloro-1,4-naphthoquinone (1b), 10 mL of dioxane:water mixture was added. The reaction mixture was irradiated with the same parameters by 15–20 min. The progress reaction was monitored by TLC, using MeOH:CHCl_3_ (Sigma-Aldrich, St. Louis, MO, USA) (9:1) as eluent mixture. Finally, 20 mL of HCl (Sigma-Aldrich, St. Louis, MO, USA) (1 N) was added and the precipitated product was filtered and purified by flash column chromatography staring with dichloromethane (DCM) (Sigma-Aldrich, St. Louis, MO, USA) and changing the polarity to finish with methanol (Sigma-Aldrich, St. Louis, MO, USA).

### 4.3. Spectroscopic Characterization of Amino Acid-1,4-Naphthoquinone Derivatives

IR, NMR, MS and EPR data are available in Appendix A. With respect to NMR characterization, some compounds reported here and in a previous work [5] presented complications due to low intensity of peaks as well as absence or presence of some protons or carbons in the spectra. This phenomenon could be explained considering that it has been reported the formation of organic radicals in some amino acid-naphthoquinone derivatives with sodium borohydride [20,21]. In this regard, we performed the EPR (electron paramagnetic Resonance) analysis of a previously reported amino acid-naphthoquinone derivative (2-((1,4-dioxo-1,4-dihydronaphthalen-2-yl)amino)-4-(methylthio)butanoic acid) [5], showing the presence of organic radicals even in the solid state (see SI). A hyperfine coupling could be observed using a basic solution (KOH 1N). Nonetheless, most analyses are required and studies on EPR and cyclic voltammetry are now in progress to explain this behavior.

2-((1,4-dioxo-1,4-dihydronphthalen-2-yl)amino)-3-(4-hydroxyphenyl) propanoic acid (3a). Orange powder. Yield: 90%. m. p.: 214–216 °C; IR (ATR): 3334.28, 2922.97, 1680.57, 1595.76, 1557.60, 1510.95, 1341.34, 1252.30, 1031.80, 777.39cm^−1^; 1H NMR (400 MHz, Methanol-d_4_) δ 7.95 (d, *J* = 8.00 Hz, 1H), 7.93 (d, *J* = 8.10 Hz, 1H), 7.70 (td, *J* = 7.60/1.20 Hz, 1H), 7.61 (td, *J* = 7.60/1.10 Hz, 1H), 7.03 (d, *J* = 8.40 Hz, 1H), 6.65 (d, *J* = 8.40 Hz, 1H), 5.55 (s, 1H), 4.09 (t, *J* = 5.40 Hz, 1H), 3.21 (dd, *J* = 13.90/4.60 Hz, 1H), 3.03 (dd, *J* = 13.90/6.80 Hz, 1H).; ^13^C NMR (75 MHz, MeOD) δ 184.69, 182.32, 177.03, 157.26, 149.17, 135.76, 134.82, 133.28, 131.87, 131.50, 129.30, 127.30, 126.76, 116.22, 100.79, 59.96, 37.88, 30.71 ppm. HRMS (ESI/Q-TOF) m/z: [M + H]+ for C_19_H_15_NO_5_: 338.0984; found: 338.1008.

2-((1,4-dioxo-1,4-dihydronaphthalen-2-yl)amino)-3-methylbutanoic acid (3b). Orange powder. Yield 92%. m. p.: 170–172 °C; (Lit: 171.1–173.1 °C). IR (ATR): 3330.04, 2922.97, 2867.84, 1680.57, 1595.76, 1566.08, 1502.47, 1341.34, 1303.18, 1053.00, 1031.80, 730.74 cm^−1^; (Lit; No rep); 1H NMR (400 MHz, Methanol-d_4_) δ = 8.09 (dd, *J* = 7.60/1.00 Hz, 1H), 8.03 (dd, *J* = 7.70/1.00 Hz, 1H), 7.79 (td, *J* = 7.60/1.40 Hz, 1H), 7.71 (td, *J* = 7.50/1.30 Hz, 1H), 5.74 (s, 1H), 5.49 (s, 1H), 3.99 (d, *J* = 5.60 Hz, 1H), 2.39–2.30 (m, 1H), 1.10 (d, *J* = 6.90 Hz, 3H), 1.05 (d, *J* = 6.80 Hz, 3H); (Lit: 1H NMR (500.13 MHz, DMSO-d6): δ = 13.22 (bs, 1H, OH) ppm; 8.01 (dd, *J* = 7.57, 0.95 Hz, 1H, CH); 7.94 (dd, *J* = 7.57, 0.95 Hz, 1H, CH); 7.85 (dt, *J* = 7.57, 1.26 Hz, 1H, CH); 7.76 (dt, *J* = 7.57, 1.26 Hz, 1H, CH); 6.85 (d, *J* = 8.51 Hz, 1H, CH); 5.72 (s, 1H, CH); 3.93–3.96 (m, 1H, CH); 2.23–2.27 (m, 1H, CH); 0.96 (d, *J* = 6.62 Hz, 3H, CH3); 0.91 (d, *J* = 6.94 Hz, 3H, CH3)_3_).; ^13^C NMR (101 MHz, MeOD) δ 185.11, 182.31, 170.32, 135.95, 133.67, 131.98, 127.47, 126.92, 101.82, 100.34, 61.83, 31.94, 19.19, 18.85 ppm; (Lit: ^13^C NMR (125.75 MHz, DMSO-d_6_): δ = 135.49 (CH), 132.98 (CH), 126.52 (CH), 125.88 (CH), 101.44 (CH), 60.67 (CH), 30.33 (CH), 19.15 (CH_3_), 19.03 (CH_3_), ppm). HRMS (ESI/Q-TOF) m/z: [M + H]+ for C_15_H_15_NO_4_: 274.1035, found: 274.1063. Characterization according to Janeczko M et al., [21].

2-((1,4-dioxo-1,4-dihidronaphthalen-2-yl)amino)-3-(1*H*-indol-3-yl)propanoic acid (3c). Orange powder. Yield 88%. m. p.: 157–159 °C (Lit: 159.1-161), IR (ATR): 3338.52, 2922.97, 1672.08, 1600, 1561.84, 1506.71, 1332.86, 1307.42, 1265.02, 828.27, 722.26; ^1^H NMR (400 MHz, Methanol-d_4_) δ 7.89 (dd, *J* = 7.70/1.30 Hz, 2H), 7.66 (td, *J* = 7.50/1.40 Hz, 1H), 7.56 (td, *J* = 4.90/2.50 Hz, 3H), 7.26 (d, *J* = 8.10 Hz, 1H), 7.10 (s, 1H), 7.00 (t, *J* = 7.30 Hz, 1H), 6.90 (t, *J* = 7.30 Hz, 1H), 5.54 (s, 1H), 4.23–4.18 (m, 1H), 3.49 (dd, *J* = 14.60/4.30 Hz, 2H), 3.12 (q, *J* = 7.30 Hz, 1H); (Lit: ^1^H NMR (300 MHz, DMSOd_6_): δ = 10.90 (s,1H), 7.92-8.00 (m,2H), 7.83 (t, *J* = 7.5 Hz, 1H), 7.74 (t, *J* = 7.5 Hz, 1H), 7.52 (d, *J* = 8.10 Hz, 1H), 7.32 (dd, *J* = 0.6, 8.1 Hz, 1H), 7.18 (d, *J* = 2.4 Hz, 1H), 7.05 (t, *J* = 7.8 Hz, 1H), 6.91–6.97 (m. 2H), 5.74 (s, 1H), 4.45–4.52 (m, 1H), 3.36-3.38 (m, 2H)).; ^13^C NMR (101 MHz, MeOD) δ 184.66, 182.27, 178.44, 149.39, 137.94, 135.70, 134.79, 133.22, 131.81, 129.05, 127.24, 126.69, 124.61, 122.32, 119.75, 119.34, 112.21, 111.20, 100.67, 59.35, 28.77 ppm; HRMS (ESI/Q-TOF) m/z: [M + H]+ for C_21_H_16_N_2_O_4_]: 361.1144, found: 361.1173. Characterization according to Janeczko et al. [21].

### 4.4. Spectroscopic Characterization of Amino Acid-2,3-Dichloronaphthoquinone Derivatives

2-((3-chloro-1,4-dioxo-1,4-dihydronaphthalen-2-yl)amino)-3-(4-hydroxyphenyl)propanoic acid (4a). Dark-red powder. Yield 95%. m. p.: 78–80 °C (Lit: 74–76 °C). IR (ATR): 3300.35, 2918.73, 2863.60, 1735.69, 1591.52, 1553.36, 1506.71, 1226.86, 1031.80, 828.27, 718.02; (Lit: IR (KBr): 3305, 1739, 1684, 1641); ^1^H NMR (400 MHz, Methanol-d_4_) δ 7.99 (d, *J* = 7.3 Hz, 1H), 7.93 (d, *J* = 7.8 Hz, 1H), 7.73 (td, *J* = 7.50/1.40 Hz, 1H), 7.65 (td, *J* = 7.50/1.30 Hz, 1H), 7.05–6.95 (m, 2H), 6.64 (dd, *J* = 23.70/8.50 Hz, 2H), 5.59 (s, 1H), 5.32 (t, *J* = 5.70 Hz, 1H), 3.14 (ddd, *J* = 38.30, 14.10, 5.80 Hz, 1H); (Lit; ^1^H NMR (300 MHz, DMSO-d_6_) δ 9.25 (br s, 1H), 7.90–7.98 (td, 2H, 7.5, 1.3), 7.82 (td, 1H, 7.4, 1.5), 7.72 (td, 1H, 7.4, 1.6), 6.91 (d, 2H, 8.4), 6.55 (d, 2H, 8.4). 5.05 (m, 1H, 5.4), 2.97–3.11 (m, 2H).; ^13^C NMR (101 MHz, MeOD) δ 184.72, 183.42, 179.78, 156.12, 134.65, 132.48, 130.46, 130.10, 125.81, 114.85, 99.79, 92.10, 58.46, 39.09 ppm. HRMS (ESI/Q-TOF) m/z: [M + H]+ for C_19_H_14_ClNO_5_: 372.0594, found: 372.0672. Characterization according to Gorohovsky et al. [23].

2-((3-chloro-1,4-dioxo-1,4-dihydronaphthalen-2-yl)amino)-3-methylbutanoic acid (4b). Red powder. Yield 92%. m. p.: 54–56 °C (Lit: 50–52 °C); IR (ATR): 3308.83, 2961.13, 2867.84, 1731.45, 1676.38, 1595.76, 1570.32, 1290.46, 1137.81, 819.79, 722.26, (Lit: IR (KBr): 3456, 3309, 1749, 1688, 1651); ^1^H NMR (400 MHz, Methanol-d_4_) δ 8.00 (dd, *J* = 7.60/1.50 Hz, 1H), 7.75 (td, *J* = 7.60/1.40 Hz, 1H), 7.67 (td, *J* = 7.50/1.40 Hz, 1H), 5.05 (d, *J* = 4.10 Hz, 1H), 2.28 (dq, *J* = 6.80/4.30 Hz, 1H), 1.26 (s, 1H), 1.12 (d, *J* = 7.00 Hz, 3H), 1.00 (d, *J* = 6.9 Hz, 3H); (Lit; ^1^H NMR (300 MHz, DMSO-d_6_) δ 8.02 (d, 1H, 75), 7.91 (d, 1H, 7.1), 7.66 (td, 1H, 7.4, 1.3), 7.53 (td, 1H, 7.5, 1.5), 6.37 (br, s, 1H), 5.01–5.12 (m, 1H), 1.03 (d, 3H, 6.9), 0.97 (d, 3H, 6.8); ^13^C NMR (75 MHz, MeOD) δ 180.22, 177.21, 174.80, 143.93, 135.07, 132.95, 130.33, 126.96, 126.41, 62.23, 32.89, 29.79, 17.70, 17.45 ppm. HRMS (ESI/Q-TOF) m/z: [M + H]+ for C_15_H_14_ClNO_4_: 308.0645, found: 308.0668. Characterization according to Gorohovsky et al. [23].

2-((3-chloro-1,4-dioxo-1,4-dihydronaphthalen-2-yl)amino)-3-(1H-indol-3-yl)propanoic acid (4c). Red powder. Yield 91%. m. p.: 148–150 °C (Lit: 153–155 °C); IR (ATR): 3423.32, 3296.11, 2927.21, 1676.33, 1595.76, 1561.84, 1404.95, 1298.94, 1260.78, 1226.86, 1031.80, 845.23, 739.22, (Lit: IR (KBr): 3437, 3318, 1728, 1680, 1602, 1567, 1511, 1342, 1300, 1272); ^1^H NMR (400 MHz, Methanol-d_4_) δ 10.28 (s, 1H), 8.01–7.84 (m, 2H), 7.62 (td, *J* = 21.20, 20.6, 10.0 Hz, 2H), 7.50 (dd, *J* = 19.00/7.60 Hz, 2H), 7.12 (d, *J* = 9.40 Hz, 1H, NH), 7.05 (d, *J* = 6.60 Hz, 1H, NH), 6.94 (t, *J* = 7.30 Hz, 1H), 6.87 (t, *J* = 7.10 Hz, 1H), 5.42 (t, *J* = 5.90 Hz, 1H), 3.48 (dd, *J* = 14.60/4.50 Hz, 2H); (Lit: ^1^H RMN (DMSO-d_6_): δ = 3.39 (m, CH2), 5.36 (m, CH), 6.58 (br, 1H), 6.93 (t, *J* = 5.60 Hz, 1H), 7.03 (t, *J* = 5.60 Hz, 1H), 7.28 (d, *J* = 6.32 Hz, 1H), 7.31 (d, *J* = 6.38 Hz, 1H), 7.49 (d, *J* = 6.32 Hz, 1H), 7.85–8.05 (m, 4H), 10.95 (NH), 12.95 (br, CO2H)); ^13^C NMR (101 MHz, MeOD) δ 179.54, 179.49, 173.75, 146.19, 136.57, 134.39, 134.17, 132.17, 127.43, 126.23, 125.59, 124.07, 123.97, 121.05, 118.56, 118.06, 110.80, 108.80, 58.01, 29.29 ppm; (Lit: ^13^C RMN (DMSO-d_6_): δ = 28.60 (CH2), 56.40 (CH), 107.9, 111.0, 111.3, 118.1, 118.5, 121.0, 124.4, 125.7, 126.4, 127.1, 129.6, 131.5, 132.8, 134.8, 136.1, 144.4, 172.6 (CO2H), 175.4 (C=O), 179.6 (C=O)); HRMS (ESI/Q-TOF) m/z: [M + H]+ C_21_H_15_ClN_2_O_4_ for 395.0754, found: 395.0770. Characterization according to Shrestha et al., [42].

### 4.5. Cyclic Voltammetry

Electrochemical studies of naphthoquinone–amino acids (3a–c) and chloride–naphthoquinone–amino acids (4a–c) derivatives were studied by cyclic voltammetry (CV) as previously reported [24]. Briefly, non-aqueous solutions of compounds 3a–c or 4a–c at a concentration of 5 mM were dissolved in dimethyl sulfoxide (DMSO, Sigma–Aldrich, St. Louis, MO, USA) containing 0.1 M tetrabutylammonium tetrafluoroborate (TBABF_4_, Sigma–Aldrich, St. Louis, MO, USA) as supporting electrolyte. Then, the non-aqueous solutions containing 3a–c or 4a–c were studied by CV in a three-electrode cell using a freshly polished glassy carbon disk electrode (GC) (0.07 cm^2^, BASi, West Lafayette, IN, USA) as working electrode and a Pt wire (Alfa-Aesar, Tewksbury, MA, USA) as counter electrode. An Ag/Ag^+^ electrode in 0.1 M TBABF_4/_DMSO was used as pseudo-reference electrode. A potential scan was performed from −2.0 V to 0.75 V vs. Ag/Ag^+^ at 100 mV s^−1^ in the cathodic direction in order to identify any electron transfer reaction associated to compounds 3a–c or 4a–c using a potentiostat/galvanostat PAR VersaStat 3 (Ametek, Inc. Berwyn, PA, USA). Furthermore, any redox reaction processes of 3a–c or 4a–c were further studied at a scan rate of 10, 25, 50, 100, 200, and 300 mV s^−1^. As received TBABF_4_ and DMSO were used without further purification or drying processes. All the non-aqueous testing solutions were degassed for 30 min with high-purity N_2_ gas before cyclic voltammetry experiments.

### 4.6. Cell Lines

HPV-16 positive, cervical tumor line SiHa; HPV-18 positive, cervical tumor line CaLo; HPV negative, cervical tumor line C33-A, and HPV negative non-tumorigenic cell line HaCaT were grown in Dulbecco’s Modified Eagle’s Medium (DMEM) (Invitrogen Corporation, Carlsbad, CA, USA) enriched with 5% fetal bovine serum (FBS). Medium change and passage were performed every 3 and 4 days, respectively. SiHa, CaLo, C33-A and HaCaT cell lines were kindly provided by Ph.D. Gariglio’s Lab from CINVESTAV-IPN.

### 4.7. Cell Proliferation Analysis

Cell proliferation was quantified by violet crystal dye in 1X phosphate-buffered saline (PBS) (2.7 mM KCl, 1.8 mM KH_2_PO_4_, 136 mM NaCl, 10 mM Na_2_HPO_4_ pH 7.4). The treated cells were incubated in methanol for 15 min and washed two times with distilled water. Cells were dyed with 0.1% crystal violet and washed three times with water and finally violet crystal was recovered with 10% acid acetic to be analyzed in microplate reader Multiskan GO Spectrophotometer (Thermo Scientific™, Ratastie, Finland).

## 5. Conclusions

As in our previous report, the synthesis of tyrosine, valine, and tryptophan naphthoquinone derivatives were obtained by microwave assisted synthesis with excellent yields and in a shorter time. Chloride naphthoquinone derivatives could be successfully prepared using a weak base TEA. The obtained results on the analysis of the synthetic methodology showed a direct dependence of the stoichiometry of the reactants and pH by deprotonation of the amino acids increasing yield with the microwave system. Electrochemical studies showed that naphthoquinones amino acid derivatives recorded a well-defined redox process and calculated redox potential E_1/2_ values give an overall insight towards their biological activity. The biological evaluation showed a potent proliferation inhibition with chloride naphthoquinone versus non-chloride amino acids derivatives. Non-chloride tryptophan (3c) and chloride Tyrosine-naphtoquinone (4a) compounds showed preferential effect in cervical tumorigenic cell lines versus immortal cell line.

## Figures and Tables

**Figure 1 molecules-25-02058-f001:**
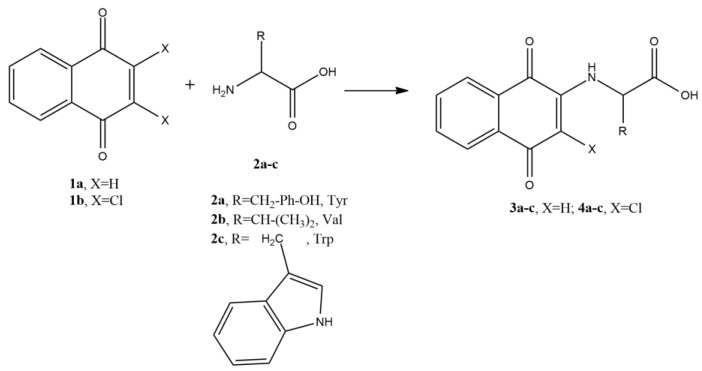
Preparation of 3**a**–**c** and 4**a**–**c** derivatives.

**Figure 2 molecules-25-02058-f002:**
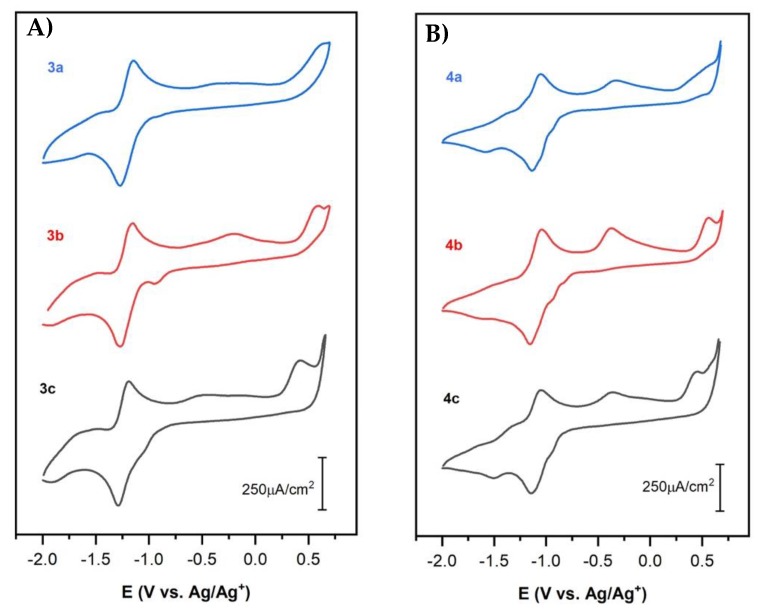
Cyclic voltammetry curves of (**A**) naphthoquinone (3a–c) and (**B**) chloride-naphthoquinone (4a–c) derivatives with a: tyrosine, b: valine and c: tryptophan at 5 mM in 0.1 M TBABF_4_//DMSO at room temperature. Scan rate: 100 mV s^−1^.

**Figure 3 molecules-25-02058-f003:**
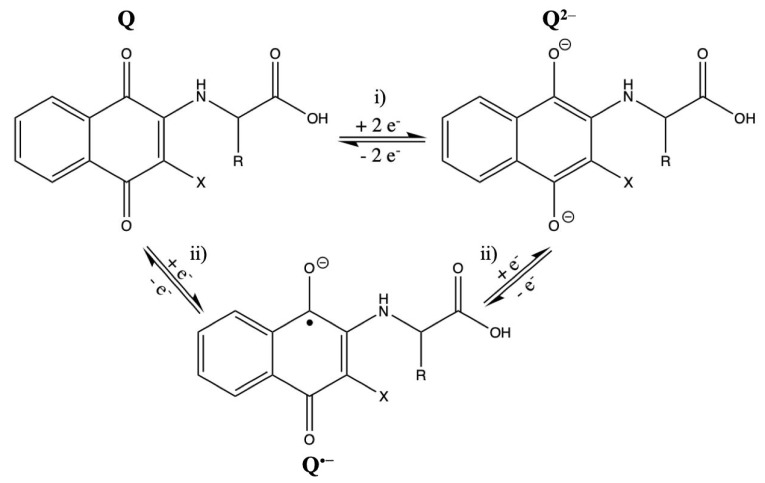
Schematic representation of the quinone moiety (**Q**) reduction in the naphthoquinone amino acid derivatives: (i) single step two-electron reduction reaction and (ii) two successive one-electron reduction reactions. X = H for 3a–c and X = Cl for 4a–c.

**Figure 4 molecules-25-02058-f004:**
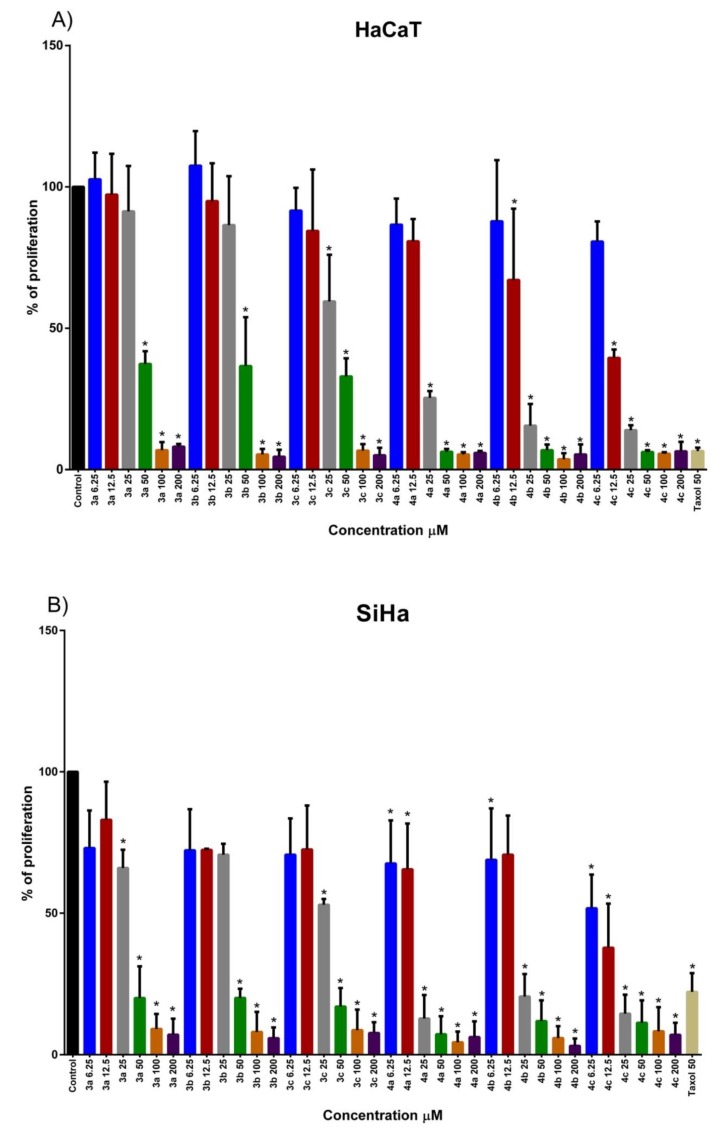
Proliferation effect of Naphthoquinone amino acid derivatives was evaluated in HPV positive cancer cell lines derived from cervix and a non-tumorigenic HPV negative cell line. (**A**) HaCaT HPV negative cell line cells were treated with 6.25, 12.5, 25, 50, 100 and 200 μM of naphthoquinone amino acid derivatives to assay proliferation rate at 72 h post-treatment. (**B**) SiHa HPV 16 positive cancer cells were treated with 6.25, 12.5, 25, 50, 100 and 200 μM of naphthoquinone amino acid derivatives to assay proliferation rate at 72 h post-treatment. (**C**) CaLo HPV 18 positive cancer cells were treated with 6.25, 12.5, 25, 50, 100 and 200 μM of naphthoquinone amino acid derivatives to assay proliferation rate at 72 h post-treatment. (**D**) C33-A HPV negative cancer cells were treated with 6.25, 12.5, 25, 50, 100 and 200 μM of naphthoquinone amino acid derivatives to assay proliferation rate at 72 h post-treatment. Cells treated with 0.1% of DMSO were used as Control. * represents statistically significant p < 0.05 value.

**Table 1 molecules-25-02058-t001:** Reaction conditions to prepare 3**a**–**c** and 4**a**–**c** derivatives.

Compound	Nq-aa	MAS ^a^ (%)	
		TEA	KOH
**3a**	1:1.8	85	90
**3b**	1:1.5	80	92
**3c**	1:1.2	80	88
**4a**	1:1.7	95	90
**4b**	1:2.5	92	87
**4c**	1:1.5	91	89

^a^ Microwave Assisted Synthesis: 110 °C, 250 W, 25 min, in dioxane-water (4:1), TEA (1mmol)/KOH (3N) 5 mL. Nq-aa: Naphthoquinone amino acid proportion.

**Table 2 molecules-25-02058-t002:** Electrochemical parameters of non-chloride and chloride naphthoquinone with tyrosine, valine and tryptophan substituents ^a^.

Compound	E_pa_	E_pc_	ΔE_p_ ^b^	E_1/2_ ^c^	i_pa_	i_pc_	|i_pa_/i_pc_|
	(V)	(V)	(V)	(V)	(mA cm^−2^)		
**3a**	−1.15	−1.27	0.13	−1.21	0.28	−0.30	0.95
**3b**	−1.16	−1.27	0.12	−1.21	0.28	−0.30	0.95
**3c**	−1.19	−1.30	0.11	−1.24	0.26	−0.27	0.95
**4a**	−1.04	−1.15	0.11	−1.10	0.22	−0.24	0.92
**4b**	−1.05	−1.19	0.14	−1.12	0.21	−0.25	0.83
**4c**	−1.05	−1.17	0.12	−1.11	0.23	−0.27	0.85

^a^ Determined by cyclic voltammetry in TBABF_4_ 0.1 M/DMSO at 100 mV/s. The potentials are given with respect to the Ag/Ag^+^ pseudo-reference electrode; ^b^ ΔE_p_ = E_pc_ − E_pa_; ^c^ E_1/2_ = (E_pa_ + E_pc_)/2.

**Table 3 molecules-25-02058-t003:** IC50 of naphthoquinone amino acid derivatives in immortal and tumorigenic cell lines after 72 h exposure. Results obtained from three independent experiments.

Cell line	Naphthoquine Amino acid Derivatives	IC50
SiIHa	3a	40.98 µM
	3b	47.22 µM
	3c	28.8 µM
	4a	6.830 µM
	4b	11.34 µM
	4c	3.209 µM
CaLo	3a	81.08 µM
	3b	56.96 µM
	3c	25.20 µM
	4a	7.028 µM
	4b	2.878 µM
	4c	2.697 µM
C33-A	3a	55.64 µM
	3b	55.56 µM
	3c	21.36 µM
	4a	~0.001577 µM
	4b	~0.0003566 µM
	4c	~0.003454 µM
HaCaT	3a	61.6 µM
	3b	55.29 µM
	3c	40.82 µM
	4a	9.882 µM
	4b	7.970 µM
	4c	~0.3311 µM

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
