# Peer review of "Differential Proliferation Effect of the Newly Synthesized Valine, Tyrosine and Tryptophan–Naphthoquinones in Immortal and Tumorigenic Cervical Cell Lines"

_molecules, 2020, doi:10.3390/molecules25092058_

Round 1
Reviewer 1 Report
The manuscript "Differential proliferation effect of the newly synthetized Valine, Tyrosine and Tryptophan–Naphthoquinones in immortal and tumorigenic Cervical Cell Lines" by Araujo-Huitrado et al. describes the synthesis and biological activity of new chloride and a achloride naphthoquinone-amino acid derivatives in view of a potential use as anticancer drugs.
The manuscript presents a series of issues that should be approached:
The use of English should be extensively revised.
The introduction is not sufficiently broad to let readers properly aproach the subject. The molecules should be better described, the literature beckground should be cited and hypothesis concerning the mechanism of action mentioned.
The crystal violet is not a very reliable method to assess cell proliferation. BrdU or at least MTT assays should be performed.
Treatments in fig. 3 should be better indicated, also using a table to make them easily readable.
The effect of the tested molecules on cancer and non cancer cells does not appear very different. Keeping to the presented results, non cancer cells appear to be more sensitive than cancer cells to treatments and this is a serious concern for any candidate drug. Experiments should be performed under the same conditions using HUVEC or normal human fibroblasts.
Author Response
Dear reviewer I respond point by point.
Comments and Suggestions for Authors
The manuscript "Differential proliferation effect of the newly synthetized Valine, Tyrosine and Tryptophan–Naphthoquinones in immortal and tumorigenic Cervical Cell Lines" by Araujo-Huitrado et al. describes the synthesis and biological activity of new chloride and a achloride naphthoquinone-amino acid derivatives in view of a potential use as anticancer drugs.
The manuscript presents a series of issues that should be approached:
Reviewer 1 observations and suggestions.
1.- The use of English should be extensively revised.
Response: The english was revised.
2.- The introduction is not sufficiently broad to let readers properly aproach the subject.
Response: The introduction was extended and improved.
3.- The molecules should be better described, the literature beckground should be cited and hypothesis concerning the mechanism of action mentioned.
Response: The molecules were depicted in a figure (Figure 1B) to achieve clarity about their structure. The background, hypothesis and mechanism of action was mentioned in the introduction.
4.- The crystal violet is not a very reliable method to assess cell proliferation. BrdU or at least MTT assays should be performed.
Response: Our group and several other researchers have used crystal violet, please check the references (Cordova-Rivas et al., Int. J Mol Sci. 2019 Jan 28;20(3). pii: E545., Rivera-Ávalos et al., Molecules. 2019 Nov 25;24(23). pii: E4285., Feoktistova et al Cold Spring Harb Protoc. 2016 Apr 1;2016(4):pdb.prot087379., Wang et al., Biochem Biophys Res Commun. 2018 Sep 26;504(1):46-53, Liu et al., Front Oncol.2020 Feb 25;10:229).
Additionally, we previously made an assay with the 4 cell lines to analyze the dynamic range of the assay by seeding 40,000 cells without treatment and analyzed them at 12, 24, 48, 72 and 96 hours.
5.- Treatments in fig. 3 should be better indicated, also using a table to make them easily readable.
Response: The treatments were indicated in figure 4 and bars are in color, and Table 3 was included with IC50 of each compound.
6.- The effect of the tested molecules on cancer and non cancer cells does not appear very different. Keeping to the presented results, non cancer cells appear to be more sensitive than cancer cells to treatments and this is a serious concern for any candidate drug. Experiments should be performed under the same conditions using HUVEC or normal human fibroblasts.
Response: We used HaCaT cells as control of cervical cancer because they are keratinocyte, which are characterized by their stratified arrangement as cervical cancer cells are. HUVEC or normal human fibroblasts normally are not used as control for cervical cancer. However, we agree about the concern that tumorigenic and immortal cells present similar effects under naphthoquinone-amino acid derivatives given the genomic alteration and accelerated proliferation in both types of cells. We know that the effect of these molecules must be further characterized analyzing other cellular and molecular processes, nevertheless, these assays are beyond the scope of the present paper.
It should be noted that we include a dose-response curve of the 6 new naphthoquines and we observed 2 compounds (3c and 4a) withwhich the effect of proliferation inhibition is lower in HaCaT than in tumorigenic cells.

Reviewer 2 Report
The manuscript entitled “Differential proliferation effect of the newly synthetized Valine, Tyrosine and Tryptophan–Naphthoquinones in immortal and tumorigenic cervical cell lines” discusses the synthesis and antiproliferation activity of six naphthoquinone-bearing amino acid derivatives. Antiproliferative activity (percent of inhibition) was carried out in SiHa, CaLo and HaCaT cells lines. In a recent publication [ref#5] the authors reported a similar study with five other amino acids (alanine, phenylalanine, methionine, glycine, and asparagine). The reviewer did not find novelty in the current study. The below revisions are recommended:
Major:
- Title: The word “synthetized” should be synthesized.
- A similar reaction should be carried out with other essential amino acids and antiproliferative activity of the products should be evaluated.
- The antiproliferative activity should be studied at a few lower concentrations as well. The studied concentration (0.1 mM) is too high.
- Give IC50 values of the six compounds against each cell line that was used during the study.
- Write the microwave and ultrasound-assisted synthetic procedures under “Materials and Methods.” A mere citation is not sufficient.
- Rewrite the IR data. A decimal (.) is missing in all the cases.
- Discussion: “…..since it is well known that the electrochemical behavior of these compounds is closely related to their therapeutic efficiency and toxic side effects [6].” The ref#6 does not support this claim. Correlate the electrochemical behavior of the six compounds with antiproliferative effects and toxicity.
- Write the coupling constants (J) up to two decimals.
- Rewrite the conclusion. It does not sound meaningful.
Minor:
- The manuscript requires a thorough revision of the language. The wried terms and abbreviations like “achloride” or “microwave (MAS) and ultrasound (UAS)” should be removed.
- Uniformity (font and size) should be maintained throughout the manuscript including the schemes and figures. References have not been cited according to the journal’s IFA (for example ref#3, 5, etc.).
Author Response
Dear reviewer I responded point by point to your observations and I attended your suggestions.
Comments and Suggestions for Authors
The manuscript entitled “Differential proliferation effect of the newly synthetized Valine, Tyrosine and Tryptophan–Naphthoquinones in immortal and tumorigenic cervical cell lines” discusses the synthesis and antiproliferation activity of six naphthoquinone-bearing amino acid derivatives. Antiproliferative activity (percent of inhibition) was carried out in SiHa, CaLo and HaCaT cells lines. In a recent publication [ref#5] the authors reported a similar study with five other amino acids (alanine, phenylalanine, methionine, glycine, and asparagine). The reviewer did not find novelty in the current study. The below revisions are recommended:
Major:
- Title: The word “synthetized” should be synthesized.
Response: The word“synthetized” was changed for synthesized.
- A similar reaction should be carried out with other essential amino acids and antiproliferative activity of the products should be evaluated.
Response: In a previous work we synthesized glycine, alanine, methionine, phenylalanine, and asparagine-naphthoquinones and were evaluated in two cell lines at one concentration, in the present work we synthesized tyrosine, valine and tryptophan-naphthoquines and we extended the study to four cell lines and we did a dose-response analysis. Additionally, we incorporated an in vitro step model of carcinogenesis using immortal and tumorigenic cells as well as the HPV status to explore in more deepness naphthoquine-amino acid derivatives effect.
- The antiproliferative activity should be studied at a few lower concentrations as well. The studied concentration (0.1 mM) is too high.
Response: We added a dose-response assay as well as an additional cell line that it is tumorigenic, HPV negative.
- Give IC50 values of the six compounds against each cell line that was used during the study.
Response: We obtained the IC50 of each compound against one cell line and they were added in Table 3.
- Write the microwave and ultrasound-assisted synthetic procedures under “Materials and Methods.” A mere citation is not sufficient.
Response: Microwave and ultrasound-assisted synthetic procedures were added in Materials and Methods.
- Rewrite the IR data. A decimal (.) is missing in all the cases.
Response: The decimal in IR data was added.
- Discussion: “…..since it is well known that the electrochemical behavior of these compounds is closely related to their therapeutic efficiency and toxic side effects [6].” The ref#6 does not support this claim. Correlate the electrochemical behavior of the six compounds with antiproliferative effects and toxicity.
Response: This part of the discussion was changed to “The overall electrochemical pathway of naphthoquinone amino acid derivatives is anticipated to be complicated since both reduction of naphthoquinone moiety and oxidation of amino acid substituents are contributing to electron-transfer reactions as reported above. Although a thorough understanding of the electron-transfer pathway is not considered in this work, the electrochemical studies of naphthoquinone amino-acid derivatives give an insight of possible biological reactions concerning electron transfer processes that could occur at cellular level. Furthermore, electrochemical parameters (Table 2) calculated by cyclic voltammetry are considered to predict the biological effect of newly synthesized compounds as reported elsewhere [33,34]. However, it is worth to mention that a direct correlation of the electrochemical parameters with biological activity can be compromised due to the complexity of biomedical chemistry or, in such a case, to the complexity of the physiological environment in living systems as compared to the media used for the electrochemical studies. In this context, the redox potential or half-wave potential (E1/2) is an electrochemical parameter that has been generally used to asses biological activity of naphthoquinone compounds [33-35]. Naphthoquinones with more positive E1/2 values are associated with higher biological effects. This is because a spontaneous reduction of the compounds occurs, thus, the generation of reactive oxygen species (ROS) is presumably facilitated. Recall that the main mechanism of actions for naphthoquinones is the generation of ROS. According to this, chloride-naphthoquinone amino-acid derivatives (4a-c) in this work are expected to present higher biological activity than the naphthoquinone derivatives without the chloride moiety (3a–c) as the E1/2 values reported in Table 2. The E1/2 values for the redox reaction of chloride-naphthoquinone amino-acid derivatives (4a-c) were found in the potential range of -1.10 V to -1.12 V meanwhile non-chloride naphthoquinone amino-acid derivatives (3a-c) exhibited values in the potential range of -1.21 V to -1.24 V. E1/2 for chloride-naphthoquinone amino-acid derivatives is, by at least, 110 mV more positive than non-chloride naphthoquinone amino-acid derivatives (3a-c). Similar results were reported by our group for chloride naphthoquinone amino acids derivatives modified with alanine, phenylalanine, methionine, glycine and asparagine suggesting that the redox reaction, as a consequence, ROS production could occur by a similar via to the naphthoquinone compounds reported in the present study [24].
- Write the coupling constants (J) up to two decimals.
Response: The two decimal were added to coupling constants.
- Rewrite the conclusion. It does not sound meaningful.
Response: The conclusion was rewritten.
Minor:
- The manuscript requires a thorough revision of the language. The wried terms and abbreviations like “achloride” or “microwave (MAS) and ultrasound (UAS)” should be removed.
Response: MAS and UAS are abbreviations described in the manuscript, therefore, we consider it is not necessary to removed. The term achloride was changed for non-chloride.
- Uniformity (font and size) should be maintained throughout the manuscript including the schemes and figures. References have not been cited according to the journal’s IFA (for example ref#3, 5, etc.).
Response: Font and size was used as template proportionate by the journal. The references were check it.
Reviewer 3 Report
I would advocate the publication of this manuscript with the following minor corrective changes:
- The authors discuss the electrochemical intermediates on page 3, but it is hard to follow this paragraph without the structures of the intermediates shown. Please provide these in a scheme or figure.
- The presentation of the proliferation data on page 5 would be better with the addition of a table including the % proliferation for each compound against each cell line plus the standard deviation. Please include.
- Please include the % proliferation of a standard compound that was discussed on page 7 (doxorubicin). Include the concentrations for it as well.
- The authors discuss hydropathy data on page 7. Do they mean cLogP data? This would be a more relevant term that would be recognized in medicinal chemistry circles. Please specify what is meant by this data and how it was gathered
- There are several issues with the experimental section. Assuming that the use of CD3OD would eliminate all exchangeable protons from the spectra, there are still protons missing from 3a and extra protons in 3b and 4c. There are also missing carbons in 4a and 4c if my calculations are correct. Please rectify this.
Author Response
Dear reviewer I responded point by point to your observations and I attended your suggestions.
Comments and Suggestions for Authors
I would advocate the publication of this manuscript with the following minor corrective changes:
- The authors discuss the electrochemical intermediates on page 3, but it is hard to follow this paragraph without the structures of the intermediates shown. Please provide these in a scheme or figure.
Response: A figure with the intermediate structures was added.
- The presentation of the proliferation data on page 5 would be better with the addition of a table including the % proliferation for each compound against each cell line plus the standard deviation. Please include.
Response: In figure 4 the concentration was specified and bars color was changed from gray to different colors bars, adittionaly, a Table was added with the IC50 of each compound.
- Please include the % proliferation of a standard compound that was discussed on page 7 (doxorubicin). Include the concentrations for it as well.
Response: The IC50 of Doxorubicin was included for breast cancer cell line.
- The authors discuss hydropathy data on page 7. Do they mean cLogP data? This would be a more relevant term that would be recognized in medicinal chemistry circles. Please specify what is meant by this data and how it was gathered.
Response: This term defines the relative hydrophobicity or hydrophilicity of an amino acid or substance.
- There are several issues with the experimental section. Assuming that the use of CD3OD would eliminate all exchangeable protons from the spectra, there are still protons missing from 3a and extra protons in 3b and 4c. There are also missing carbons in 4a and 4c if my calculations are correct. Please rectify this.
Response: This phenomenon could be explained considering that it has been reported the formation of organic radicals in some amino acid-naphthoquinone derivatives. We proved this with EPR analysis. A paragraph about this explanation was added in the spectroscopic characterization section.
Round 2
Reviewer 1 Report
The manuscript has been considerably improvedby the Authors and is now suitable for publication in Molecules.
Reviewer 2 Report
The authors have made some improvements in their revised manuscript.